# Regulatory T Cell-Based Adoptive Cell Therapy in Autoimmunity

**DOI:** 10.3390/ijms262110340

**Published:** 2025-10-23

**Authors:** Eduardo Gozálvez, Adrián Lario, Guillermo Muñoz-Sánchez, Francisco Lozano

**Affiliations:** 1Servei d’Immunologia, Centre de Diagnòstic Biomèdic, Hospital Clínic de Barcelona, 08015 Barcelona, Spain; gozalvez@clinic.cat; 2Servei de Dermatologia, Hospital Clínic de Barcelona, 08015 Barcelona, Spain; 3Immunoreceptors of the Innate and Adaptive System, Institut d’Investigacions Biomèdiques August Pi I Sunyer, 08015 Barcelona, Spain; 4Department de Biomedicina, Universitat de Barcelona, 08015 Barcelona, Spain

**Keywords:** autoimmunity, regulatory T cells (Tregs), engineered T cell receptor (eTCR), chimeric antigen receptor (CAR)

## Abstract

Regulatory T cells (Tregs) are a distinctive subset of CD4^+^ T cells critical in self-tolerance maintenance to prevent the development of autoimmunity. The mechanisms by which these cells provide immune regulation are numerous and, consequently, deeply involved in the pathogenesis of many autoimmune disorders. Treg-based adoptive cell transfer (ACT) therapy has generated interest as a novel, promising strategy to restore self-tolerance in autoimmunity. Polyclonal Treg-based ACT therapy was first implemented in clinical trials, presenting adequate safety profiles. Subsequent preclinical studies have shown antigen-specific Tregs to be safer and more effective than polyclonal approaches, so research has recently moved in this direction. Antigen-specificity can be conferred to Tregs by viral transduction of genes coding for engineered T cell receptors (eTCRs) or chimeric antigen receptors (CARs), with encouraging outcomes in different animal models of autoimmunity. This review focuses on the biology of Tregs, as well as on current preclinical and clinical data for Treg-based ACT in the field of autoimmunity.

## 1. Introduction

Immune tolerance refers to the ability of the immune system to remain unresponsive to self-antigens. When this capacity is disrupted, immune responses targeting the body’s own tissues arise, leading to what we know as autoimmune disease. Among the different cells that aid in the maintenance of this tolerance, a specialised naturally occurring subset termed CD4^+^ regulatory T cells (Treg) are central for the prevention of autoimmunity [1,2,3].

These autoimmune and autoinflammatory disorders pose a major burden to healthcare systems as they affect approximately 5–10% of individuals in industrialised countries [3,4,5,6]. Their heterogeneity and complexity have hampered the development of targeted therapies, as treatment must adequately suppress autoreactivity while preserving the normal functional side of the immune response [2,7,8]. Historically, first-line therapies have relied on broad immunosuppression with corticosteroids, cytotoxic drugs such as azathioprine or methotrexate, or disease-modifying antirheumatic drugs (DMARDs) like leflunomide, hydroxychloroquine, and sulfasalazine. Nonetheless, these approaches are limited by their non-specific mechanisms, chronic use, and significant side effects, which include a higher risk of infections and malignancy [5,9,10].

Biological drugs, particularly monoclonal antibodies (mAbs) directed against immune mediators such as TNF-α, have represented important advances in the treatment of some autoimmune conditions. Still, these drugs do not restore immune tolerance, and patients frequently experience treatment failure [5,11]. Consequently, there is an unmet need for strategies that actively re-establish self-tolerance [7].

In this context, adoptive cell therapy (ACT) using Treg cells has gained considerable attention as a promising therapeutic avenue in autoimmunity. In recognition of the centrality of regulatory T cells (Tregs) to immune homeostasis, the 2025 Nobel Prize in Physiology or Medicine honoured Mary E. Brunkow, Fred Ramsdell and Shimon Sakaguchi for discoveries that established FOXP3^+^ Tregs as key enforcers of immune tolerance. This milestone crystallised the concept that immune regulation is an active process essential to preventing autoimmunity and maintaining balance, providing the conceptual foundation for the Treg-based adoptive cell therapies discussed in this review [12,13,14,15].

## 2. Treg Historical Perspective

In the 1970s, Gershon and Kondo [16] identified a population distinct from helper T cells that could dampen the immune responses. These cells were initially termed “suppressor T cells”. Subsequent studies suggested heterogeneity within this population, with some subsets acting in an antigen-specific manner and others functioning more broadly [17]. However, methodological challenges and inconsistent results led to scepticism in the field, causing the “suppressor T cell” concept to fall out of favour [18,19]. At the same time, tolerance research advanced through the recognition of clonal deletion and anergy as key mechanisms [20,21,22].

Parallel experimental approaches focused on tolerance breakdown and the induction of autoimmunity. Nishizuka and Sakakura [23] demonstrated that neonatal thymectomy in mice triggered autoimmune ovarian destruction and the related production of autoantibodies. Later, Penhale et al. [24] showed that thymectomy in adult rats induced autoimmune thyroiditis, an effect that could be prevented by transferring CD4^+^ T cells from healthy syngeneic donors [25,26]. These findings indicated that the thymus gives rise to a subset of CD4^+^ T cells with suppressive functions, suggesting the coexistence of effector and regulatory CD4^+^ populations [27].

The next step taken was to separate these two CD4^+^ populations in normal naïve mice by the expression of cell surface molecules [28,29,30,31,32,33]. These attempts identified CD25 (the IL-2 receptor α-chain) as a key feature. Removal of CD25^+^ CD4^+^ cells induced severe autoimmune disease in mice, whereas reconstitution with this subset prevented disease onset [29]. This provided evidence for a distinct population of regulatory CD25^+^ CD4^+^ T cells, later designated regulatory T cells (Treg) [19].

A major breakthrough came in the early 2000s with the discovery of the transcription factor FoxP3 as the master regulator of Treg development and function [13,34,35]. The link between FOXP3 mutations and IPEX syndrome, inflammatory bowel disease, and severe allergy further underscored its essential role in immune tolerance [36,37,38].

It is now well established that Tregs are indispensable for controlling aberrant responses against self, microbial, and environmental antigens. Their dysfunction can contribute to autoimmune and allergic diseases, while their suppressive properties also facilitate tumour immune evasion, posing challenges for cancer immunotherapy [39,40].

## 3. Treg Biology

### 3.1. Treg Development, Modulation, and Types

Treg account for roughly 5–10% of circulating CD4^+^ T cells in peripheral blood (PB). The most widely accepted classification distinguishes between thymus-derived or natural Treg (tTreg/nTreg) and peripherally induced Treg (pTreg) [Table 1] [8].

nTreg arise in the thymus during positive selection of CD4^+^ single-positive thymocytes and are committed to a lineage that is functionally and phenotypically distinct from other T cell subpopulations [39,41,42]. This lineage commitment is not accomplished by simple positive selection but rather through an agonist-selection programme that depends on relatively strong self-reactive TCR signals delivered in the thymic medulla [43,44]. Within this framework, TCR signal strength operates within a window of outcomes: low/tonic signals support conventional positive selection of Tconv, very high-affinity signals trigger clonal deletion, and intermediate-to-high ‘agonist’ signals—integrated with key cytokine and co-stimulatory cues—divert cells to the Treg lineage [44]. IL-2–STAT5 signalling is indispensable for Foxp3 induction and stabilisation, with STAT5 engaging Foxp3 regulatory elements and scaling the size of the tTreg niche [45,46,47]. CD28 co-stimulation is required for tTreg development, acting in concert with TCR signals and IL-2 [48], while ICOS–ICOSL interactions fine-tune selection and support Treg differentiation/maintenance in the thymic environment [49,50]. Antigen availability is provided by AIRE- and FEZF2-dependent programmes in medullary thymic epithelial cells (mTECs) and by dendritic cells that acquire and present tissue-restricted antigens; these antigen-presentation axes coordinate deletion versus Treg diversion of strongly self-reactive clones, shaping the emergent Treg repertoire and tissue specificity [51,52,53]. Collectively, these data establish agonist selection—rather than simple positive selection—as the defining pathway for thymic Treg generation, integrating TCR signal strength with IL-2/STAT5 and CD28/ICOS co-stimulation in the context of medullary antigen presentation [43,44,45,54]. Together, these convergent signals initiate FOXP3 transcription and the early stabilisation of the Treg lineage programme, providing a mechanistic bridge from thymic selection to lineage commitment.

Building on this selection-driven signalling, the development and function of nTreg is coordinated by a transcription factor network involving FoxP3 and the establishment of a specific epigenetic landscape [14,41]. The Treg epigenome encompasses Treg-specific DNA demethylation, reinforcing expression activation and stabilisation of super-enhancers associated with Treg gene signature (FOXP3, CD25, CTLA-4, HELIOS, EOS) [2,14]. Signals downstream of IL-2/STAT5 and TCR/co-stimulation converge on chromatin to promote Treg-specific DNA demethylation, thereby consolidating the lineage programme. Within this framework, the FOXP3 CNS2 region contains a conserved CpG island (Treg-specific demethylated region, TSDR) that is hypomethylated (transcriptionally active) in nTreg but is hypermethylated and inactive in Tconv. This hypomethylated TSDR region is considered the most reliable indicator of a stable Treg lineage, as it ensures high FoxP3 levels [39].

Although FOXP3/Foxp3 expression and the Treg epigenome normally confer lineage stability, inflammatory cytokines and strong co-stimulation can bias Treg towards functional plasticity. In particular, IL-6–STAT3 signalling antagonises Treg programmes and favours Th17-like features; under sustained inflammation, some Tregs down-regulate Foxp3 and produce IL-17A or IFN-γ, a phenomenon reported in human disease and experimental models. These observations underscore that Treg identity is not immutable in hostile cytokine milieus [55,56,57].

At the molecular level, FOXP3 cis-regulatory elements, notably CNS2/TSDR, are crucial to maintain high FOXP3 expression and lineage stability [58,59]. Demethylation of the TSDR and coordinated activity of FOXP3 enhancers safeguard the Treg programme, whereas inflammatory cues that erode this landscape predispose to instability [58,59]. Experimental perturbation confirms causality: targeted TSDR demethylation stabilises FOXP3 in human T cells; conversely, loss of CNS2 function destabilises the lineage [60,61].

Beyond FOXP3, additional transcription factors (e.g., BACH2, IKZF family/Eos, C/EBP) and pathways (PI3K–AKT–mTOR) modulate the balance between suppressive and effector programmes, providing rational entry points for therapeutic interventions aimed at preserving Treg function in inflammatory settings [60,62].

In contrast, pTregs differentiate in peripheral tissues, like the intestinal mucosa or the maternal placenta, from naïve CD4^+^ T cells upon antigen stimulation in environments rich in TGF-β, IL-2, and retinoic acid [8,39]. Unlike nTregs, which primarily recognise self-antigens, pTregs often respond to non-self-antigens from commensal microbiota, food, or allergens, helping maintain mucosal and peripheral tolerance [2,41].

Additional subsets with regulatory properties, also induced in the periphery, are Th3 and Tr1 cells. Th3 cells, first defined by Weiner, emerge in the gut following oral antigen exposure and produce TGF-β and IL-4 [61], while Tr1 cells, as termed by Cottrez and Groux, arise under IL-10-rich conditions and are defined by high IL-10 secretion with minimal IL-2 and IL-4 production [63,64]. In contrast to nTregs, Th3 and Tr1 subsets contribute to tolerance toward food, microbial, and even some-self antigens [65,66,67,68].

Besides these naturally occurring Treg subtypes, a third group of regulatory cells that are artificially induced in vitro under non-inflammatory conditions could be established; these are termed “induced Tregs” (iTregs) [69]. These in vitro-induced regulatory cells are likely similar to their pTreg counterparts, as their induction from FoxP3^-^ T cells is achieved through incubation conditions that resemble those that promote pTreg generation [70,71,72]. It is noteworthy that these iTreg may exhibit greater instability in inflammatory environments, raising questions about their therapeutic applicability [73].

While CD4^+^ Treg cells have been extensively studied, their CD8^+^ counterparts remain far less understood [74], although their potential contribution to the therapy field of autoimmunity makes them an ideal candidate to explore. Evidence indicates that CD8^+^ Treg are a heterogeneous group with diverse phenotypes, origins, and mechanisms of action. Among them, FoxP3^+^ CD8^+^ CD28^low^ CD8^+^, CD103^+^ CD8^+^, and CD8αα^+^ CD4^−^ T cells have been described [75,76,77,78,79,80,81].

FoxP3^+^ CD8^+^ Treg cells are rare under steady-state conditions in both mice and humans [82], but they can expand under pathological circumstances such as transplantation, autoimmunity, or cancer [74,83,84,85,86,87]. Their developmental origin is not fully resolved, though recent studies suggest they may be generated peripherally rather than exclusively in the thymus [88]. As with CD4^+^ Treg, their induction relies on TCR signalling along with co-stimulatory and cytokine-mediated signals, particularly IL-2 and TGF-β.

Functionally, CD8^+^ Treg suppress immune responses through both contact-dependent and cytokine-mediated mechanisms. They can inhibit effector T cell activity via high IL-2 consumption, production of IL-10, TGF-β, and IFN-γ, or by competing for γ-chain cytokines [89]. Additionally, some CD8^+^ Treg subsets display cytotoxic properties, resembling conventional CD8^+^ effector T cells, although the relevance of this function in vivo remains uncertain [74].

### 3.2. CD4^+^ Treg-Mediated Suppression

Treg employ a broad repertoire of suppressive strategies [Figure 1], including direct cell–cell interactions, secretion of immunomodulatory cytokines, and modulation of the microenvironment [41].

A defining feature of Treg cells, especially upon activation, is their inability to produce IL-2 due to FoxP3-driven transcriptional repression. Instead, they depend on exogenous IL-2 for survival, which they efficiently capture via constitutive expression of the high-affinity IL-2 receptor (CD25). By consuming IL-2, Treg deprive conventional T cells (Tconv) of this cytokine, limiting their proliferation and differentiation into effectors [41].

Another major mechanism involves cytotoxic T-lymphocyte antigen 4 (CTLA-4), expressed constitutively by Treg. CTLA-4 binds with high affinity to CD80/CD86 on antigen-presenting cells (APCs), preventing CD28-mediated co-stimulation of effector T cells. Additionally, CTLA-4 removes CD80/CD86 from the APC surface through transendocytosis and induces indoleamine-2,3-dioxygenase (IDO) in APCs, which depletes tryptophan and further inhibits effector T cell activity [41].

Treg also secrete anti-inflammatory cytokines such as IL-10, TGF-β, and IL-35, which directly suppress effector T cells. In addition, they can mediate cytotoxicity via granzyme/perforin-dependent mechanisms [9,41]. Expression of the ectoenzymes CD39 and CD73 enables the breakdown of extracellular ATP into adenosine, which binds A2A receptors on effector cells, exerting suppressive effects [90].

Beyond direct inhibition, Treg can act indirectly by fostering the development of additional regulatory populations, a phenomenon referred to as “infectious tolerance”. Once activated by specific antigens, Treg can exert bystander suppression over effector T cells of different specificities, extending their regulatory influence beyond the initiating antigen [91,92].

Interestingly, Tregs have also been implicated in tissue repair and regeneration. They can produce growth factors such as amphiregulin, keratinocyte growth factor, and neuregulin-1, contributing to regeneration in muscle, lung, intestine, and other tissues [93]. Furthermore, Treg cells modulate neutrophil survival and cytokine production, protect against inflammation-induced tissue damage, and influence processes such as angiogenesis and metabolic homeostasis [72,93,94,95].

## 4. Treg in Human Autoimmune Disease

### 4.1. IPEX Syndrome and IPEX-like Monogenic Diseases

Mutations of Treg signature genes (FOXP3, IL2R, and CTLA4) impair Treg development and function, resulting in severe autoimmunity [2,96].

FOXP3 mutations produce intense Treg-specific dysfunction, resulting in IPEX syndrome, which is a rare early-onset condition characterised by multiorgan autoimmunity. IPEX syndrome is often fatal in the first years of life and typically includes the triad: neonatal-onset type 1 diabetes (T1D), eczematous dermatitis, and enteropathy [96]. To date, only long-term immunosuppression or bone marrow transplantation have been effective therapies, carrying significant side effects. However, novel approaches, including targeted gene editing of FOXP3, are being investigated as potential treatment options [97].

Similarly, CD25 deficiency and CTLA-4 haploinsufficiency can lead to severe autoimmune syndromes (IPEX-like syndromes), which reconfirm the important role of CTLA-4 and CD25 expression in Treg function [2].

### 4.2. Treg in Multifactorial and Polygenic Autoimmune Diseases

Common polygenic autoimmune diseases, such as T1D or multiple sclerosis (MS), afflict 5–10% of the population worldwide and are caused by chronic immune responses against the host’s cells, subsequently culminating in tissue damage. These autoimmune disorders develop from interactions between environmental triggers and genetic polymorphisms that induce a loss of self-tolerance [9].

Since the discovery of Treg indispensable role in self-tolerance, efforts have been made to detect Treg abnormalities in autoimmunity [8,39]. In the last decade, an increasing number of papers reported either functional or numerical anomalies regarding Treg population in multiple autoimmune diseases, including T1D, MS, autoimmune hepatitis (AIH), rheumatoid arthritis (RA), vitiligo, and IBD, among others [98,99,100,101,102].

The observed Treg defective function in autoimmune diseases involves their inability to inhibit Teff activation and proliferation, as well as Teff secretion of inflammatory cytokines [98,100]. The decreased capacity of Treg in regulating autoreactive cells depends on many factors, but their downregulated FoxP3 protein expression (Treg instability) seems to be uniformly present [39,103].

## 5. Treg-Based Therapies in Autoimmunity

Treg’s crucial role in maintaining self-tolerance, and therefore, controlling autoimmune responses, reveals the clinical potential of Treg-based therapies. They can find broad applications in restoring self-tolerance in autoimmune disorders and inducing tolerance to allogeneic tissues upon solid organ and hematopoietic stem cell transplantation. Treg-based therapies are being investigated with promising outcomes in the transplantation field [90], but this review will only discuss their application in autoimmunity.

Novel therapeutic approaches include either ACT of ex vivo expanded Treg or immunomodulatory interventions to enhance Treg expansion in vivo. Both applications can potentially promote Treg-mediated immune regulation in a non-specific (polyclonal) or antigen-specific way [Figure 2] [9,104].

In vivo immunomodulatory interventions were first attempted in the treatment of AIDs. They enhance Treg-mediated function in a polyclonal or antigen-specific manner according to the mechanism of action targeted. On one side, the polyclonal approach includes interventions that promote Treg-specific IL-2 signalling [105] (e.g., low-dose IL-2, mutant IL-2 and IL-2/anti-IL-2 mAb complexes), interventions that preferentially inhibit pathogenic T cells over Treg (e.g., anti-CD3 mAbs and mTOR inhibitors) or activation of Treg-co-stimulatory receptors (e.g., tumour necrosis factor receptor 2, TNFR2), or even gut microbiome transplantation [72,106,107]. On the other side, the antigen-specific approach comprises tolerogenic dendritic cell-based therapies (e.g., antigen-specific DCs engineered to overexpress 1α-hydroxylase in combination with non-toxic zinc concentration [108] or antigen-specific tolerogenic vaccines (e.g., protein/peptide- and DNA/RNA-based vaccines) [7,9,104,109,110,111]. This review only focuses on Treg-based ACT.

## 6. Treg-Based ACT in Autoimmunity

The therapeutic potential of Treg-based ACT in autoimmunity has been demonstrated in multiple preclinical models of autoimmune disorders, which have promoted advances in their clinical development, with polyclonal Treg therapy most widely used in clinical trials [92]. Nowadays, around 14 clinical trials (active and completed) have tested the safety and efficacy of Treg-based ACT in different autoimmune diseases [Table 2]. There are currently four main products developed for Treg-based ACT: polyclonal Treg, endogenous antigen-specific Treg, antigen-specific induced Treg (iTreg), and engineered Treg (eTreg) [Figure 2] [92,104,112].

### 6.1. Polyclonal Treg Therapy

Treg can be isolated from either peripheral blood (PB) or umbilical cord blood (UCB). Those from PB have a low precursor frequency, making them difficult to extract and expand effectively [119]. In addition, peripheral Treg cells are unstable in PB and can be transformed into Teff cells. In contrast, Treg cells from UCB of allogeneic donors have a higher percentage of naïve T cells, maintaining a highly activated state with immunosuppressive function [119]. They also have a unique homing capacity, ideal in ACT for autoimmune disorders [119]. A third promising source besides PB and UCB could be neonatal thymus, as they are often discarded in paediatric cardiac surgery and contain more undifferentiated Treg cells than UCB and PB, which, indeed, have a stronger suppressive capacity [120]. Nevertheless, this source needs to be further investigated [3,8].

Despite the potential of UCB and allogeneic sources, current Treg-based ACT predominantly utilises expanded autologous Treg from PB because of its accessibility [116]. Two primary methods: magnetic-activated cell sorting (MACS) and fluorescence-activated cell sorting (FACS). In the first one, from the isolated mononuclear cell population via the standard gradient centrifugation technique, further purification is needed to sort out desired cells with immunosuppressive capacities. This is achieved through the employment of a magnetic field that distinguishes and separates cells attached to magnetic beads. In this sense, there is a first step of CD8^+^ and CD19^+^ cell depletion (double negative selection) followed by enrichment of the CD25^+^ fraction (positive selection). This MACS-based method is in accordance with the specifications of GMP production, as it works in a closed system, but its accuracy is suboptimal, reaching only 80% of FoxP3^+^ cells in the final product [5,8,90,92,121].

Instead, another advanced method to isolate Treg is a flow cytometry-based purification (FACS), where Treg differentiation is based on fluorescent labelling. Therefore, it is possible to select this population according to CD4^+^ CD25^high^ CD127^low^ expression. This ensures a superior separation purity (>99%), but the technique is limited by the slower processing speed and the inability to guarantee aseptic operation, making it difficult to meet the requirements of GMP production [8,114,119,122]. Nevertheless, novel closed and GMP-compliant cell sorting systems, such as the MACSQuant^®^ Tyto^®^ (Miltenyi Biotec, Bergisch Gladbach, Germany) or the CGX10 Cell Isolation System (Sony Biotechnology), have recently been developed to enable sterile, clinical-grade FACS-based Treg isolation. Although the resulting cells are relatively pure after isolation, the existence of contaminating Teff cells may outgrow Tregs over time [123,124]. Thus, an additional selection of the naïve (CD45RA^+^) population can enhance purity by eliminating activated T cells that transiently upregulate CD25 [124]

Secondly, post-Treg-isolation, the cell count is usually insufficient; thus, expansion in cell cultures is required to achieve optimal concentration for therapeutic applications, making it the primary challenge of Treg therapies. Polyclonal Treg isolated are usually expanded ex vivo using high-dose IL-2 and anti-CD3/CD28 mAb-coated beads [125,126]. It is important to bear in mind that a high purity of the initial Treg cell population ensures to increase in Treg yield, and some researchers have studied their suppressive effect. The incorporation of rapamycin into the cell culture is especially relevant when the Treg population has been purified by MACS, as it helps to compensate for the considerably reduced initial purity [112]. Nonetheless, it has been shown [127] that rapamycin is not needed when Treg cells are purified using FACS methods with selection of CD4^+^ CD25^hihg^ CD127^low^ Treg cells. It is noteworthy that the process of expanding Tregs not only expands the number of Tregs but also has been shown to correct impairments of Treg function [114,128,129]. Finally, polyclonal Treg are reinfused intravenously as an autologous Treg-based ACT [8,90,92,130].

The first data on the use of polyclonal Tregs was focused on graft-versus-host disease (GvHD) in 2009 [122] and Crohn’s disease in 2012 [131]. Now, polyclonal Treg therapy has been or is currently being tested in patients with T1D, AIH, IBD, systemic lupus erythematosus (SLE), systemic sclerosis (SS), pemphigus, and Guillain–Barré syndrome [Table 2]. Moreover, completed phase I clinical trials in T1D, MS, IBD, and SLE have revealed that autologous polyclonal Treg therapy is well tolerated and safe [114,115,116,117,118,119].

Particularly in T1D, where most studies have been carried out, recent phase l clinical trials have assessed and proved the safety and feasibility of polyclonal Treg therapy to reverse recent T1D onset [114,119,120,132]. Specifically, Bluestone et al. found [114] that these expanded polyclonal Tregs maintained their Treg cell state and polyclonality, and they could be detected in circulation after a year post-transfusion. A similar follow-up phase I study, followed by IL-2 administration, was confounded by the non-Treg effects of IL-2 [120]. Marek-Trzonkowska et al. [132] also demonstrated safety with possible efficacy in a phase I new-onset T1D study with a limited number of children. Nonetheless, the small cohort sizes and differences in the age and disease stage of the treated cohorts may have contributed to the variability in results between these studies [119].

The safety and efficacy of polyclonal Treg therapy were also tested in the first and recent randomised placebo-controlled phase II clinical trial (Sanford Project T-Rex Study, NCT02691247) [133]. This was a large, well-powered study to evaluate a range of Treg doses in a blinded manner, including a placebo group. It was found that better C-peptide preservation was achieved when the cells presented a lower in vitro fold expansion. On the other hand, higher doses of cells with a higher in vitro fold expansion did not correlate with better outcomes. Hypothetically, with lower in vitro fold expansion, Tregs are more functional in vivo or may have an enhanced ability to expand further in vivo. It is difficult to speculate whether the mixed results respond to variances in patients and study design (e.g., different dosages) or if the therapy is inefficient in its current setting. Thus, these findings highlight the need to better understand Treg expansion biology and consider new avenues for the improvement of current ACT Treg therapies [119].

### 6.2. Polyclonal vs. Antigen-Specific Treg Therapies

As previously mentioned, various clinical trials have been performed employing polyclonal Treg therapy, providing relevant lessons on production procedures; cell dosages; cell circulation in patients; and Tregs’ stability, safety, tolerability, and clinical efficacy (e.g., prolonged C-peptide levels in T1D trials) [1,114,118]. Nevertheless, the current difficulties in building improved nTreg ACT include antigen specificity, functional stability, and survivability [105].

Various preclinical models of organ transplantation and autoimmune diseases show that antigen-specific Tregs enriched by in vitro antigen stimulation are superior to polyclonal Tregs [105,134,135,136,137]. Antigen-specific Treg predominantly localise at the site of antigen presentation, reducing the risk of generalised/off-target immunosuppression and making them both more effective and safer than polyclonal Treg for ACT [104,138]. In addition, the increased trafficking of antigen-specific Treg to inflamed target tissues probably permits the infusion of lower cell numbers [92,104]. For all these reasons, in the last years, research has moved towards the development of antigen-specific approaches to treat autoimmune disease, but no clinical trials using such ACT Tregs are ongoing in the autoimmunity field.

The purification and expansion of endogenous disease-relevant autoantigen-specific Tregs remain technically challenging owing to their very low frequency in PB, as most of them are localised in the target tissue, limiting their accessibility. The endogenous antigen-specific Treg approach cannot be adapted for the treatment of autoimmune diseases [92,104]. Antigen-specific Treg has been assessed to prevent graft rejection in transplantation using APC from the graft donor to specifically stimulate alloreactive-specific Treg from the recipient in vitro, producing a differentiation into effector-type nTregs (CD45RA^lo^ CD25^hi^ FoxP3^hi^), which are Bcl-2^lo^ and prone to die by apoptosis upon TCR re-stimulation. This could explain why the transfer of a large number of antigen-specific nTregs generally does not incur general immunosuppression [105,139].

Thus, alternative strategies are needed to produce antigen-specific Tregs for a particular antigen in the autoimmunity field, such as the generation of antigen-specific Treg in vitro by transformation of antigen-specific Teff into cells with suppressive capacity (antigen-specific iTreg cells) or redirecting polyclonal Treg by genetic insertion of synthetic antigen receptors (engineered Treg cells, eTreg) [104].

### 6.3. Antigen-Specific Induced Treg Therapy

FoxP3^+^ Treg can be induced from Teff in vitro, provided they are stimulated under conditions of high IL-2 and TGF-β levels, generating in vitro iTreg [39]. This could have the advantage over nTregs of in vitro preparation of a large number of antigen-specific iTregs from CD4^+^ Tconvs, serving as a potential source to produce antigen-specific iTregs for ACT [104,105]. By definition, Teff-derived (or Tconv-derived) iTregs retain the native T cell receptor (TCR) of the precursor cell; therefore, their antigen specificity is dictated by that original TCR. As a consequence, polyclonal Teff-to-iTreg conversions yield a repertoire-wide mixture of specificities mirroring the diversity of the starting TCR pool, whereas induction from antigen-enriched precursors produces oligoclonal or monoclonal iTregs with focused specificity.

However, in contrast to nTreg, it has become clear that iTregs are phenotypically and functionally more unstable, especially under inflammatory conditions. This is clinically relevant as iTreg may regain their Teff features in vivo and enhance autoimmune response. This instability of TGF-β-induced iTreg could be mainly explained because of their incomplete Treg-type epigenomic changes at FOXP3 [39,105,140]. Nevertheless, it has been shown that deprivation of the CD28 co-stimulatory signal at an early stage of iTreg generation can establish Treg-specific DNA hypomethylation at Treg signature genes, allowing production of stable antigen-specific iTreg [105,141].

Other strategies to reprogram Teff into Treg include inhibition of cyclin-dependent kinases (CDK) 8/19 signalling pathways in combination with TGF-β [141,142], the use of ascorbate (vitamin C) that could facilitate de novo hypomethylation of Foxp3 CNS2 in iTregs through Tet activation [143,144], the use of epigenetic regulators (e.g., inhibition of HDAC6 [145,146,147]), and transgenic overexpression of the transcription factors FOXP3, HELIOS and BACH2 via lentiviral vectors or lately via more advanced genetic tools (e.g., CRISPR-Cas9 genome editing systems) [97,142,148]. Indeed, enforced expression of FOXP3 on conventional CD4^+^ T cells establishes an effectively stable Foxp3^+^ Treg phenotype and associated suppressive functions, even under inflammatory conditions [5,97,148,149]. It has been seen that the combination of several of these methods can be applied. For example, the combined use of an anti-CD3, a CDK8/19 inhibitor, and TGFβ resulted in a 10-fold increase in CD25^high^ FoxP3^+^ Tregs in cultured CD4^+^ T lymphocytes compared to unstimulated lymphocytes due to the transdifferentiation of antigen-specific effector T lymphocytes into Tregs [150].

All these approaches could confer immunosuppressive functions to Teff, preserving their Treg-like properties in vivo when transferred into different autoimmunity preclinical models, but their clinical utility in common autoimmune diseases must be determined [105,142,148]. Indeed, a registered clinical trial (NCT05241444), which is still recruiting patients, aims to test in 30 participants with IPEX syndrome the infusion of autologous CD4^+^ T cells, which have undergone lentiviral-mediated gene transfer of the healthy human FOXP3 gene [151].

### 6.4. Engineered Treg Therapy

Another approach to produce antigen-specific Treg is to redirect polyclonal Treg by genetically introducing, through lentiviral/retroviral vectors, synthetic receptors that recognise a target antigen of interest. These receptors can take the form of engineered TCRs (eTCRs) or chimeric antigen receptors (CARs) [Figure 3]. New strategies to generate these engineered Treg consist of FOXP3 genetic transduction in combination with the introduction of eTCRs or CARs to convert Teff into immunosuppressive antigen-specific Treg-like cells [152,153]. In most preclinical and translational studies, however, CAR-/TCR-modified Tregs are derived from natural Tregs (CD4^+^ CD25^+^ CD127^low^ FOXP3^+^) that are isolated and then genetically engineered, whereas Teff-to-Treg conversion via FOXP3 plus a CAR/eTCR remains an alternative, more experimental route [154].

On one hand, eTCRs are produced through the integration of genes that code for TCR α and β chains specific for an antigen of interest. This heterodimeric nature of TCRs makes them complex to engineer, as the introduced subunits may recombine with the endogenous TCRα or TCRβ chains. This difficulty raises the risk of generating TCRs with unknown specificities and off-target effects [112]. Numerous strategies have been envisaged to overcome this problem. One is the insertion of extra cysteine residues in the constant region of eTCRα and eTCRβ chains to form a disulfide bond between the two to promote preferential pairing of the transduced chains [155,156]. Another strategy is the replacement of the human constant region in TCR chains with a mouse constant region, as interspecies pairing of TCR chains has never been observed [157,158]. Nonetheless, the current most promising approach is to apply genome editing tools to engineer the endogenous TCR locus. This could be achieved through knocking out one or both endogenous TCR chains or, more elegantly, by the replacement of the endogenous TCR with the new TCR by knocking it into the TCRα constant region locus (TRAC) [159].

Given their effectiveness and safety in cancer, CARs are considered an encouraging strategy for the clinical administration of antigen-specific Treg in autoimmune disorders [160,161]. CARs are formed by the artificial fusion of different protein motifs, and basically involve 3 major components: (1) an extracellular domain, commonly derived from the single chain variable fragment (scFv) of a mAb that specifically recognises the desired antigen in an MHC-independent manner; (2) a transmembrane domain; and (3) an intracellular domain, which grants the signal transduction for cell activation. Diverse CAR generations have been described, differing in the number/type of intracellular domains. However, the second-generation CAR has been the most used in CAR-Treg designs, especially the one that integrates the co-stimulatory molecule CD28 as an intracellular domain [162], which seems to be essential for potent CAR-Treg functions. The optimal design of CAR-Treg remains under investigation; however, tailored signalling architectures may be required to accommodate differences in the affinity of, and the propensity for self-aggregation amongst, scFv [112]. Separately, it should be noted that CD19-directed CAR-T cells used to treat autoimmune disease are typically conventional T cells rather than Tregs, and multiple early clinical reports (including case series and prospective cohorts) have shown high rates of remission in refractory conditions such as SLE, supporting the concept of B-cell depletion/immune reset without the need for a Treg phenotype [163,164,165].

The pioneers of CAR-Treg research were Megan Levings et al. They developed a CAR specific to HLA-A2 to prevent rejection of solid grafts. The experiments showed that Treg cells with the A2-CAR construct retained their phenotype and ability to suppress the immune response before, during, and after stimulation, thereby proving the concept of using CAR-Tregs in immune-mediated responses [72,166]. After this solid proof, CAR-Tregs have been used in the treatment of various disease models [Table 3]. Excitingly, the first-in-human study of a genetically engineered Treg product (NCT04817774) is currently ongoing as part of the STEADFAST phase I/II trial to test CAR-Tregs specific for HLA-A2 in renal transplantation [167]. Encouragingly, intermediate results from this study showed that CAR-Tregs were well tolerated in the 6 patients dosed across 4 dose levels [168].

Likewise, TCRs have been used to generate TCR-Tregs for the treatment of various AID models [Table 3] such as T1DM, MS, acquired factor VII deficiency… [91,177,178,179] and have shown their efficacy. Interestingly, a recent study demonstrated the possibility of developing Sm-specific TCRs and the production of Sm-specific Tregs for the treatment of SLE patients positive for anti-Sm and HLA-DR15. In this work, Sm-Tregs effectively suppressed inflammatory responses and inhibited disease progression in a humanised mouse model of lupus nephritis [180].

Although preclinical evidence has supported the development of eTCR- and CAR-Treg therapies [Table 3], these engineered receptors hold different functional features that could represent advantages/disadvantages depending on the context. While CARs present a greater affinity for their specific antigen than eTCRs, CARs require a higher density of antigen to trigger their activation. Hence, eTCR may be more useful for clinical settings where the disease-relevant antigen is lowly expressed, whereas CARs are probably more useful in autoimmune disorders associated with highly expressed antigens on target tissue and low expression on normal tissues [92,161]. Furthermore, antigen recognition by eTCR-Treg is MHC-restricted and thereby only recognises pMHC complexes and requires matching of the patient’s MHC genotype. MHC restriction and the necessity to identify/isolate antigen-specific and disease-relevant TCRs may limit the translation of eTCR-Treg into the clinic and may be the reason why CARs are more widely studied [104]. Instead, CAR-Treg can recognise soluble/surface antigens in an MHC-independent manner, enabling the specific detection of antigens of diverse nature (whole proteins, glycolipids, gangliosides…) and expanding their application to a larger number of patients (no matching of MHC genotype required) [9,161].

## 7. Challenges and Future Perspectives on the Field of Treg Therapies

### 7.1. Challenges in TCR-Treg-Based Therapies

A key challenge in developing eTCR-Treg therapies is the identification of suitable target antigens that are expressed on specific HLA alleles [72,176,181]. In this sense, there are three major determinants. First, it is necessary to explore whether TCRs from Tregs or Teff cells are more promising sources for their isolation [182]. Second, it may be necessary to identify those TCRs that are restricted by widely expressed HLA molecules. Third, sequence data from both TCRα and TCRβ will be needed to determine TCR specificity [72,112].

Another limitation in choosing the proper TCR is selecting the affinity with which it will bind to the pMHC complex. In this regard, both low- and high-affinity TCRs have been compared in studies, which have shown that they can be positively [177] and negatively [152] correlated with immunosuppressive function. Nonetheless, this association is dependent on a more complex pattern. For instance, Sprouse et al. showed [183] that TCRs with high- and low-affinity TCRs migrated to the pancreas, though executing differential functions. The ones with high affinity were demonstrated to activate classical pathways (CTLA-4, IL-10, etc.), and the ones with low affinity expressed proteins associated with tissue repair.

### 7.2. Challenges in CAR-Treg-Based Therapies

One restraint of CAR-Treg is their ability to accomplish suppressor functions by interacting with tissue-specific autoantigens [184]. Therefore, if these antigens are expressed not only in the site of an autoimmune reaction but also in other healthy tissues, the immune response may be less effective and lead to systemic hyperactivation of CAR-Treg that, in turn, can cause a nonspecific immunosuppressive reaction [185]. On this basis, several modifications of the classical CAR-Treg product have been proposed and studied to overcome these limitations.

An optimisation of the CAR-Treg product to overcome this problem could be the manufacturing of universal CAR-Tregs (UniCAR-Tregs) [72]. These are designed based on universal tumour-targeted CAR-T cells (UniCAR-T), which are two-component systems. The first component is a universal CAR-T that, instead of interacting with human surface antigens, interacts with a concrete peptide motif. This motif is then contained in the second component, a soluble adapter called the targeting module (TM) [186]. TM are bispecific molecules that link UniCAR-T/Treg cells to target cells. Thus, UniCAR-T/Treg could be turned on and off by dosing the TM [187], ensuring the maintenance of the CAR-T/Treg as inert until they encounter the appropriate target module. As far as we are aware, two groups have tested these constructs in Tregs, proving their high versatility and easy reprogrammability [145,188].

In addition to UniCARs, there are numerous other CAR constructions that are currently being investigated, each of which offers its unique benefits (Figure 4). Since Grada et al. [189] first developed the idea of dual targeting cancer cells using bispecific TanCARs, the application of this concept to Treg to Tregs has gained attention. Another potential strategy could be the replacement of the scFv of the CAR with the antigen itself to regulate B-cells that are antigen-specific, which is known as B-cell-targeting Ab receptors (BARs). Furthermore, to target combinations of antigens that are widely expressed throughout the body, the incorporation of SynNotch receptors may prove valuable. These receptors incorporate a two-step positive feedback circuit that comprises a priming signalling receptor devoid of activating-signalling capacity, which, upon antigen recognition, releases a transcription factor, thus initiating the expression of a CAR targeting another antigen [145].

The idea of addressing these diseases in an antigen-specific way is attractive, but there remains the paramount obstacle of the search for the concrete antigen, where in many clinical scenarios, the pathogenic antigen is unknown, or multiple antigens are involved, with the potential for epitope spreading [190,191]. The exploration of new targets will greatly benefit these cellular therapies.

One way to address the above limitation is targeting a T cell activation-associated antigen, resulting in an engineered Treg product capable of suppressing a wide range of autoimmune disorders [192]. As demonstrated by the group of Rui X. [191], a prime candidate target may be OX40L (CD252), a TNF superfamily protein expressed on activated DCs, B cells, and endothelial cells [193], which constitutes a critical co-stimulatory signal for T cell activation through binding to its cognate receptor OX40 (CD134) during antigen presentation and helps in the development of T follicular helper (Tfh) cells and autoreactive B cells [194,195,196]. Even though this group developed the cell product for the GvHD caused by stem cell transplantation, the fact that OX40L is overexpressed in RA [197,198], systemic sclerosis [199,200], and SLE [194,201,202], makes it and apparently ideal target to consider in future investigations.

This notion of targeting inflammatory mediators for the production of CAR-Tregs was further developed in other studies [154,203], and the concept of a CAR was converted into what is known as artificial immune receptors (AIRs). AIR-Treg development may help to solve the problem of choosing antigens for Treg-based therapies, as these inflammation biosensors are targeted at inflammatory mediators, including TNF-α, TNF-like ligand 1A… instead of antigens [203]. The structure of an AIR may resemble that of a CAR as they are composed intracellularly of a CD3ζ chain and a CD28 co-stimulatory signalling domain, and extracellularly by the receptor to the inflammatory ligand [72].

### 7.3. Challenges in Engineering Strategies of Artificial Receptors and Treg Functionality

A critical specific limitation of eTCR and CAR approaches in Treg-based ACT is the current use of viral vectors to insert the receptor transgene. Apart from manufacturing complexities, viral vectors imply random transgene integration, which carries the risk of oncogenic changes. Instead, the arrival of non-viral genome editing technologies, such as CRISPR-Cas9, has enabled alternative strategies to potentially improve the safety, efficacy, and manufacturing of genetically modified antigen-specific T cells [92]. In addition, novel episomal vector systems have been developed in which the incorporation of a scaffold matrix attachment region (S/MAR) into the vector DNA ensures persistent mitotic stability, resistance to epigenetic silencing, and efficient expression of the gene of interest [204,205]. Recently, human CAR-T cells were generated by nano-nS/MARt vectors, and they mediated efficient tumour killing [154,206]. Nonetheless, application in Tregs and clinical studies has yet to be carried out.

In addition to antigen specificity via eTCR/CAR, Treg cell migration into disease-specific inflamed sites could be reinforced by transgenic introduction of chemokine receptors, which might restrict their activation to target tissue, thereby limiting the risk of off-target activity. For instance, patients suffering from MS may benefit from Treg expressing CXCR3, as studies have suggested that T cells require its expression to brain homing [207].

Insights into Treg plasticity under inflammatory conditions argue for product and regimen designs that actively promote FOXP3 stability and resist Th1/Th17 deviation. First, source selection matters: CD45RA^+^ (naïve) nTregs display superior epigenetic stability and a lower propensity to acquire effector cytokines compared with memory-phenotype Tregs, which translates into more reliable expansion and function after infusion [208,209]. Second, ex vivo culture conditions can be tuned to preserve lineage commitment: IL-2–STAT5 support together with mTOR inhibition (rapamycin) maintains FOXP3 and suppressive function during expansion, while excessive PI3K–AKT–mTOR signalling correlates with loss of the Treg programme [209,210]. Approaches that favour FOXP3-CNS2/TSDR demethylation—including vitamin C/TET-facilitation in iTreg settings—further consolidate lineage stability [60,211]. Third, co-interventions in vivo can buffer hostile cytokine milieus: low-dose IL-2 preferentially amplifies Tregs via STAT5 and has shown acceptable safety across multiple autoimmune indications, whereas IL-6/STAT3 pathway blockade mitigates Treg-to-Th17 drift in inflamed tissues [212,213,214]. Taken together, stability-aware sourcing, manufacturing, and adjunct therapy can be systematically combined to increase the odds of durable Treg engraftment, function, and clinical benefit in autoimmune settings.

Regarding engineered Treg therapies’ safety, CAR-Treg seem to be likely safer than conventional CAR-T therapies, as it has been proven that they do not elicit tumour lysis syndrome, haemotoxicity, or cytokine release syndrome [145]. Nonetheless, there are some risks. One potential off-target effect could result from their immunosuppressive effects, such as an increased risk of infection or cancer. However, all phase I/II clinical trials using Treg ACT have demonstrated the safety of this therapy [215]. Another likely concern would be the contamination of Treg products with Tconv cells, exacerbating the underlying autoimmune or alloimmune disease [145]. In addition, some in vitro studies reported that CAR-stimulated Treg might exhibit cytotoxic activity, which needs to be deeply studied to avoid unexpected side effects [216]. To resolve these theoretical issues, one approach used for conventional CAR-T therapy could be translated into the CAR-Treg field. This is the implementation of an “OFF” switch to eliminate CAR-Treg without disturbing the immune system. That could be achieved by the utilisation of a kinase inhibitor, such as dasatinib, or an enzyme inhibitor, to reversibly lead to CAR degradation [145,217]. Moreover, for eTCR/CAR-Treg therapies, genetic approaches may also be used to induce Treg expression of suicide cassettes, which can be activated if an adverse event occurs (e.g., cell instability, off-target toxicity, immune suppression), allowing rapid elimination of the engineered Treg [92,218].

## 8. Manufacturability and Scalability of Treg Therapies

The production of autologous Treg-based cell therapies continues to face significant challenges that limit their application beyond rare disorders such as acute lymphoblastic leukaemia and multiple myeloma. Expanding these therapies to more prevalent autoimmune diseases, including systemic lupus erythematosus and multiple sclerosis, introduces considerable industrial, logistical, and financial hurdles [219].

While the cell therapy industry has made progress in establishing standardised practices for commercial CAR-T production, the generation of Tregs remains in its early stages. This is largely due to their low abundance in circulation and the lack of a unique surface marker to facilitate their selective isolation, making Tregs among the most technically demanding cell types to manufacture [220,221].

Across autologous therapies, three major challenges persist: scalability, dose determination, and cost. Scaling up Treg production is particularly difficult, as current processes are labour-intensive and rely on open operations with highly specialised equipment. Developing automated, integrated, and closed manufacturing systems will be essential to improve efficiency and patient throughput. Additionally, accurate dose generation is critical, since Tregs proliferate more slowly than CAR-T cells in vivo, necessitating sufficient cell yields for therapeutic effect. Finally, the high cost of cell therapies—driven by single-use reagents, specialised equipment, skilled labour, and extensive analytical testing—remains a key barrier, and addressing these factors will be important for broader clinical accessibility [220].

What is more, patients with autoimmune conditions often require long-term treatment, which can adversely affect T cell numbers and function, further complicating the collection and production of high-quality autologous Treg products. In this context, allogeneic, “off-the-shelf” cell therapies derived from universal donors may offer a viable solution to overcome the inherent limitations of autologous approaches [222].

## 9. Conclusions

Treg cells play a crucial role in the maintenance of self-tolerance. The mechanisms by which they provide immune regulation are multiple and complex, and their dysfunction has been demonstrated to contribute to the pathogenesis/development of multiple autoimmune diseases, but they also take part in other processes, such as graft rejection or cancer immune evasion mechanisms.

Treg-based ACT has generated enthusiasm for the treatment of autoimmune disorders, where current treatments often alleviate symptoms and promote episodes of remission but do not cure the disease. Instead, Treg-based ACT is believed to provide a potential cure for autoimmunity by durably establishing tolerogenic cell populations to eliminate autoreactivity.

Polyclonal Treg therapy was the first-implemented Treg-based ACT in human clinical trials, showing correct safety profiles and limited efficacy. However, several preclinical studies have revealed that antigen-specific Tregs, including engineered Tregs (CARs or eTCRs), are safer and more effective. Thus, research has moved towards the development of antigen-specific approaches in recent years, with encouraging outcomes in multiple preclinical studies.

Nevertheless, to develop safe and effective antigen-specific Treg therapies, further studies of human Treg biology in physiological and pathological conditions are required, applying new approaches to characterise different Treg subsets, as well as better strategies to identify disease-relevant autoantigens and Treg anomalies. All of this, together with the current efforts to develop useful tools to engineer Tregs, will be essential to enhance and optimise Treg-based ACT.

## Figures and Tables

**Figure 1 ijms-26-10340-f001:**
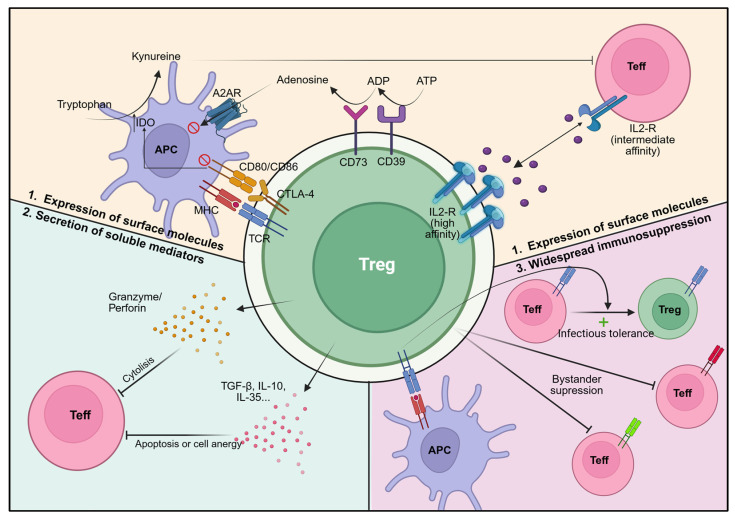
Treg-mediated suppression mechanisms. Foxp3^+^/FOXP3^+^ Treg (green) suppress Teff (red) and other immune cells through multiple, non-mutually exclusive pathways: (1) Expression of surface molecules: CTLA-4, which binds to CD80/CD86 on antigen presenting cells (purple), leading to IDO expression and tryptophan deprivation, which is essential for Teff function; CD39 and CD73, which mediate adenosine release via conversion of ATP and ADP, respectively; CD25, which allows expression of high-affinity IL-2R and IL-2 deprivation from Teff cells; (2) Secretion of soluble factors: immunosuppressive factors such as anti-inflammatory cytokines (TGF-β, IL-10, and IL-35) or cytolytic factors such as granzyme and perforin release; (3) Regulating other T cells: Treg activation via antigen-specific recognition induces suppressive effects on Teff of different specificities (bystander suppression), as well as propagation of their immune regulatory properties to neighbouring cells (infectious tolerance). Created with BioRender.com (accessed on 1 July 2025).

**Figure 2 ijms-26-10340-f002:**
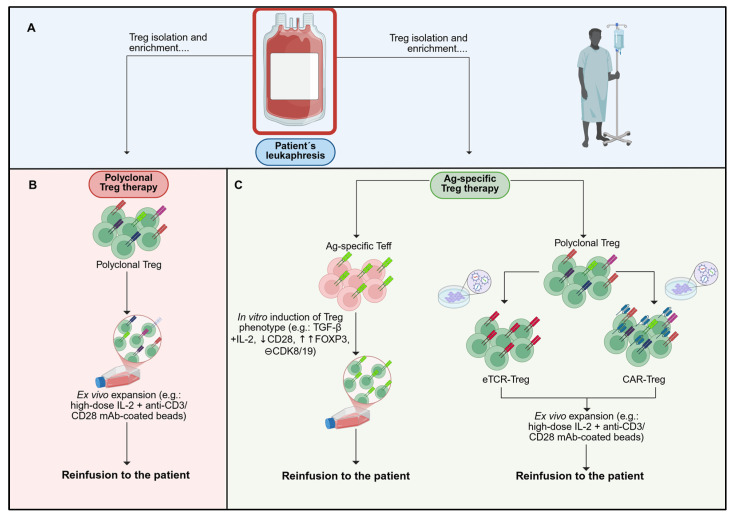
Schematic representation of polyclonal and antigen-specific Treg manufacturing workflows. Current Treg-based ACT products can be categorised by antigen specificity. (**A**) Isolation of Treg is typically based on surface markers (CD4^+^ CD25^high^ CD127^low^), optionally enriching the naïve subset (CD45RA^+^), using magnetic selection or FACS. (**B**) Polyclonal Treg products retain a broad TCR repertoire, reflecting that of the starting Treg population. Following isolation, ex vivo expansion is performed to achieve clinically relevant doses while preserving the regulatory phenotype and function. (**C**) Antigen-specific products are generated via two principal routes: (i) selection of antigen-enriched T cell precursors (e.g., Teff/Tconv bearing a defined TCR) followed by induction of a Treg phenotype under appropriate conditions (e.g., FOXP3-directed conversion); or (ii) genetic redirection of isolated Treg to express synthetic antigen receptors, namely engineered TCRs (eTCR; MHC-restricted, peptide–MHC recognition) or chimeric antigen receptors (CAR; MHC-independent), conferring specificity to a defined target antigen. Colour/key: panel A (isolation), panel B (polyclonal/blue), panel C (antigen-specific/orange). Created with BioRender.com (accessed on 1 July 2025).

**Figure 3 ijms-26-10340-f003:**
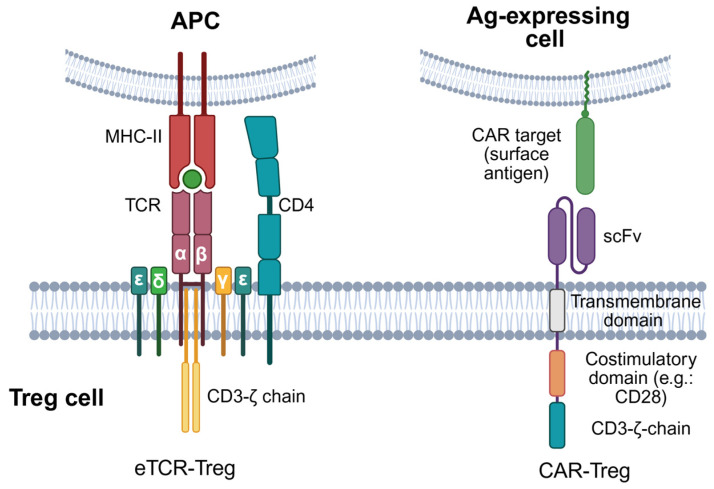
Synthetic antigen-specific receptors in engineered Treg therapy. Antigen-specific Treg can be engineered by transducing genes encoding synthetic TCRs targeting antigenic peptides in the context of MHC class I or class II (pMHC-I or -II) molecules, or CARs enabling direct antigen recognition in a pMHC-independent manner. Engineered TCRs, same as endogenous TCRs, are composed of TCRα and TCRβ chain heterodimers binding to specific pMHC-I/II complexes and associated with a CD3 complex composed of three signalling subunits: one CD3ζ homodimer, one CD3εδ heterodimer, and one CD3γε heterodimer. CARs are hybrid antigen receptors made up of an extracellular scFv capable of binding a target antigen; a transmembrane domain; and an intracellular part, including activating motifs from CD3ζ and co-stimulatory receptors (e.g., CD28 or 4-1BB/CD137), with CD28 being the most current used co-stimulatory domain in CAR-Treg. Created with BioRender.com (accessed on 1 July 2025).

**Figure 4 ijms-26-10340-f004:**
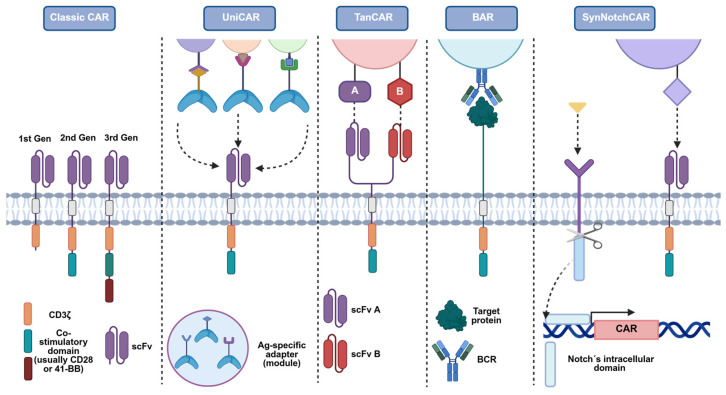
Design variants of the classical CAR to address antigen-specificity and control constraints in CAR-Treg therapy. Schematic comparison of modular receptor architectures built on the classical CAR backbone (scFv–hinge–transmembrane–co-stimulatory domain–CD3ζ). Each format offers distinct advantages and trade-offs for specificity, controllability, and safety in Treg applications. Classic CAR: single scFv confers fixed, MHC-independent recognition of a defined surface antigen. UniCAR: a two-component system in which cells express a “universal” CAR that recognises a short peptide epitope; adapter molecules (bearing that epitope and an antigen-binding moiety) retarget the cells to different antigens and enable on/off, dose-tunable control. TanCAR: a bispecific receptor with two extracellular scFv, allowing recognition of two antigens (A and B) and the possibility of combinatorial (OR/AND-like) targeting to improve precision. BAR/CAAR: the scFv is replaced by the autoantigen itself to engage autoreactive B-cell receptors (BCRs) on pathogenic B-cell clones, enabling selective recognition and potential deletion of disease-relevant B cells. SynNotch-CAR: a two-step circuit in which a ligand (yellow)-specific SynNotch receptor (purple) signalling-deficient for activation induces de novo CAR expression upon sensing a first cue, thereby restricting CAR activity spatially/temporally to sites where both cues are present. Figure adapted from Ref. [145]. Created with BioRender.com (accessed on 1 July 2025).

**Table 1 ijms-26-10340-t001:** Differential properties and characteristics of the different Treg subtypes.

Property	tTreg	pTreg	iTreg	Th3	Tr1	CD8^+^ Treg
Development	Thymus	Periphery	In vitro	Mucose	Periphery	Periphery
Progenitor cell	CD4 single positive	Naïve CD4^+^ T cells	Naïve or conventional T cell	Naïve CD4^+^ T cells	Naïve CD4^+^ T cells	Naïve CD8^+^ T cells
Antigen specificity	Autoantigens	Bacterial and food antigens	Non- and self-antigens	Bacterial flora, pathogens, food, and self-antigens	Bacterial flora, pathogens, food, and self-antigens	Non-self-antigens
Ex vivo expansion	Anti-CD3 and IL-2	-	-	-	-	-
In vitro differentiation	-	-	Anti-CD3, anti-CD28, IL-2 and TGF-β	-	-	-
TSDR methylation status	Demethylated	Unstably demethylated	Methylated	Demethylated	Demethylated	-

**Table 2 ijms-26-10340-t002:** Completed and ongoing clinical trials of Treg-based adoptive cell therapy in autoimmunity as published on clinicaltrials.gov, clinicaltrialsregister.eu, and isrctn.com (accessed between 1 May and 20 September 2025).

Disease	Study ID	Phase	Enrolment (Age)	Product	Treg Dose *	Status (Ref.)
T1D	ISRCTN6128462	I	12 (5–28 y.o.)	Expanded autologous (PB) polyclonal Treg	10 and 30 × 10^6^ cells/kg	Completed [113]
T1D	NCT01210664	I	16 (18–45 y.o.)	Expanded autologous (PB) polyclonal Treg	0.05, 0.4, 3.2, and 26 × 10^8^ cells/kg	Completed [114]
T1D	NCT02691247	II	13 (8–17 y.o.)	Expanded autologous (PB) polyclonal Treg	2.5 and 20 × 10^6^ cells/kg	Completed
T1D	NCT02772679	I	16 (18–45 y.o.)	Expanded autologous (PB) polyclonal Treg + IL-2	3 and 20 × 10^6^ cells/kg	Completed [115]
T1D	NCT02932826	I/II	40 (6–60 y.o.)	Expanded UCB polyclonal Treg	2 × 10^6^ cells/kg	Recruiting
T1D	NCT03011021	I/II	40 (>18 y.o.)	Expanded UCB polyclonal Treg + liraglutide	2 × 10^6^ cells/kg	Unknown
T1D or LN	NCT05566977	I	20 (>18 y.o.)	Expanded autologous (PB) polyclonal Treg	N/A	Unknown
SLE	NCT02428309	I	1 (46 y.o.)	Expanded autologous (PB) polyclonal Treg	1 × 10^8^ cells	Completed [116]
Pemphigus	NCT03239470	I	5 (18–75 y.o.)	Expanded autologous (PB) polyclonal Treg	1 and 2.5 × 10^8^ cells	Completed
SS	NCT0524014	I/II	30 (18–80 y.o.)	Expanded autologous (PB) polyclonal Treg	N/A	Completed
AIH	NCT02704338	I/II	30 (10–70 y.o.)	Expanded autologous (PB) polyclonal Treg	10–20 × 10^6^ cells/kg	Unknown
Crohn’s disease	NCT03185000	I/II	24 (18–80 y.o.)	Expanded autologous (PB) polyclonal Treg	N/A	Unknown
UC	NCT04691232	I	10 (18–75 y.o.)	Expanded autologous (PB) polyclonal Treg	0.5, 1, 2, and 10 × 10^6^ cells/kg	Completed [117]
MS	EudraCT:2014-004320-22	I/II	14 (18–55 y.o.)	Expanded autologous (PB) polyclonal Treg	40 × 10^6^ cells/kg (IV) or 1 × 10^6^ cells/kg (IT)	Completed [112,118]

Abbreviations: AIH, autoimmune hepatitis; GBS, Guillain-Barré syndrome; IL-2, interleukin 2; IT, intrathecal administration; IV: intravenous administration; LN, lupus nephritis; MS, multiple sclerosis; N/A, not available; T1D, type 1 diabetes; Treg, regulatory T cells; SLE, systemic lupus erythematosus; SS, systemic sclerosis; UC, ulcerative colitis; y.o., years old. * Terminated due to participant recruitment feasibility.

**Table 3 ijms-26-10340-t003:** CAR- and eTCR-based Treg therapies in preclinical models of autoimmune diseases.

Disease	Animal Model	T Cell Subset Used	Synthetic Receptor	Target Antigen	Delivery Route	Outcomes	Ref.
Colitis (IBD)	TNBS-induced colitis in BALB/c or C57BL/6 mice	Mouse CD4^+^ CD25^+^ Treg	2nd-generation CAR (Co-stim: CD28)	TNP	Retroviral	Amelioration of TNBS-induced colitis	[169]
Colitis (IBD)	Teff-mediated colitis or AOM-DSS-induced colitis in CEA BAC mice	Mouse CD4^+^ CD25^+^ Treg	2nd-generation CAR (Co-stim: CD28)	CEA	Retroviral	Enhanced migration to colon and amelioration of Teff-mediated colitis	[134]
MS	MOG-induced EAE in HLA-DR15 transgenic mice	Human CD4^+^ CD25^hi^ CD127^lo^ Treg	HLA-DR15-restricted engineered TCR	MBP	Retroviral	Suppression of Teff proliferation and amelioration of EAE	[170]
MS	MOG-induced EAE or PLP-induced EAE in 2D2 C57BL/6 mice	Mouse CD4^+^ CD25^+^ FoxP3^+^ Treg	I-A^b^-MGC II-restricted engineered TCR	MOG or MOG/NF-M	Retroviral	Bispecific eTCR-Treg had a superior capacity to monospecific, and exerted bystander suppression	[91]
MS	MOG-induced EAE in C57BL6/mice	Mouse CD4^+^ T cells + lentiviral transduction of FOXP3	2nd-generation CAR (Co-stim: CD28)	MOG	Lentiviral	Increased migration to the brain and amelioration of EAE	[135]
MS	MOG-induced EAE in C57BL/6 mice	Human CD4^+^ CD25^hi^ CD127^lo^ Treg	2nd-generation CAR (Co-stim: CD28)	MOG or MBP	Retroviral	Significant reduction of EAE disease score and suppression of progression	[171]
Vitiligo	Transgenic h3TA2 mouse model of spontaneous vitiligo	Mouse CD4^+^ FoxP3^+^ Treg	2nd-generation CAR (Co-stim:CD28)	GD3	Retroviral	Increased IL-10 secretion, superior cytotoxicity, control, and delay in depigmentation	[172]
RA	mBSA-induced arthritis in C57BL/6 mice	Mouse CD4^+^ CD25^+^ Treg and CD4^+^ T cells + retroviral transduction of FOXP3	I-A^b^-MHC II-restricted engineered TCR	OVA	Retroviral	Suppression of mBSA-induced arthritis	[173]
RA	CIA mouse model	Human Treg	CAR (generation not provided)	CV	Not provided	CV-CAR-Treg expanded when cultured with synovial fluid from patients with RA	[174]
T1D	NOD and NOD BDC2.5 mice	Human CD4^+^ T cells by combining FOXP3 gene homology-directed repair editing	HLA-DR0401-restricted engineered TCR	Islet specific antigens (IGRP, GAD65 or PPI)	Lentiviral	Homing to the pancreas and blockade of diabetes induction.	[152]
T1D	NOD/Ltj mice	Mouse CD4^+^ T cells + retroviral transduction of FOXP3	2nd-generation CAR (Co-stim: CD28)	Insulin	Retroviral	Persistence of the cells in diabetic mice, although not preventing diabetes.	[153]
T1D	Humanised T1D mouse model	Mouse Treg	CAR (generation not provided)	GAD65	Not provided	Increased pancreatic localisation, large Treg population in pancreas and spleen, and lowered glucose levels.	[175]
T1D	In vitro	Human CD4^+^ T cells, CD8^+^ T cells and CD4^+^ CD25^+^ CD127^lo^ Treg	2nd-generation CAR (Co-stim:CD28)	HPi2	Retroviral	CAR-Treg failed to maintain expansion due to tonic signalling. HPi2-CAR cannot be exploited.	[176]

## Data Availability

No new data were created or analyzed in this study. Data sharing is not applicable to this article.

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
