# Peer review of "Regulatory T Cell-Based Adoptive Cell Therapy in Autoimmunity"

_ijms, 2025, doi:10.3390/ijms262110340_

Round 1

Reviewer 1 Report

Comments and Suggestions for Authors

This is a comprehensive review on the applications of Tregs in autoimmune diseases, albeit a bit lengthy. The following concerns should be addressed to avoid confusion for readers:

1) In section 6.1 (e.g. lines 271 - 280): Please clarify whether the expanded cord blood–derived Tregs are autologous or allogeneic.

2) In lines 301 - 304: Please revise the wording to make it clear that GMP-grade cell sorting is available.

3) In lines393 - 394: Please expand the sentence and explain the antigen-specificity of Teff-derived iTregs. 

4) In section 6.4: Please clarify whether the TCR/CAR-modified Tregs are derived from Teff or natural Tregs. In addition, briefly note that CD19-directed CAR T cells have shown high efficacy in treating autoimmune diseases (e.g., SLE) and that they do not necessarily need to be Tregs.

Author Response

We thank the reviewer for the careful evaluation of our manuscript and for the constructive suggestions provided. We have revised the text accordingly. All changes are tracked in the re-submitted files and highlighted in the manuscript.

Comment 1

In section 6.1 (e.g. lines 271–280): Please clarify whether the expanded cord blood–derived Tregs are autologous or allogeneic.

Response 1
We appreciate this helpful comment. We have now explicitly clarified that umbilical cord blood (UCB)–derived Tregs are an allogeneic source in the context discussed. We initially assumed this would be evident, as an autologous UCB product would imply a neonatal recipient—an uncommon scenario for cell therapy infusion—but we agree the clarification improves readability.
Updated text inserted:
“In contrast, Treg cells from UCB of allogeneic donors have a higher percentage of naïve T cells, maintaining a highly activated state with immunosuppressive function” (line 269, Section 6.1)

Comment 2

In lines 301–304: Please revise the wording to make it clear that GMP-grade cell sorting is available.

Response 2
Thank you for pointing this out. We agree that the original wording could be interpreted as if GMP-compliant FACS were not available. We have revised the paragraph to state clearly that closed, GMP-compatible FACS platforms exist, and we added representative examples. (Indeed, our Immunotherapy Laboratory currently operates such equipment.)
Where: Section 6.1, FACS paragraph (revised text; highlighted in track changes).

Updated text inserted:
“Nevertheless, novel closed and GMP-compliant cell sorting systems, such as the MACSQuant® Tyto® (Miltenyi Biotec) or the CGX10 Cell Isolation System (Sony Biotechnology), have recently been developed to enable sterile, clinical-grade FACS-based Treg isolation” (lines 299-302)

Comment 3

In lines 393–394: Please expand the sentence and explain the antigen-specificity of Teff-derived iTregs.

Response 3
We agree this clarification is important. We have added a concise explanation stating that Teff- (or Tconv-) derived iTregs retain the native TCR of the precursor cell, which dictates their antigen specificity; thus polyclonal conversions mirror the breadth of specificities present in the starting TCR repertoire, whereas antigen-enriched precursors yield focused (oligo/monoclonal) specificity.
Where: Section 6.2 (iTreg generation and properties), paragraph detailing iTreg biology (revised text; highlighted in track changes).

Updated text inserted:
“By definition, Teff-derived (or Tconv-derived) iTregs retain the native T-cell receptor (TCR) of the precursor cell; therefore, their antigen specificity is dictated by that original TCR. As a consequence, polyclonal Teff-to-iTreg conversions yield a repertoire-wide mixture of specificities mirroring the diversity of the starting TCR pool, whereas induction from antigen-enriched precursors produces oligoclonal or monoclonal iTregs with focused specificity” (lines 386-391).

Comment 4

In section 6.4: Please clarify whether the TCR/CAR-modified Tregs are derived from Teff or natural Tregs. In addition, briefly note that CD19-directed CAR T cells have shown high efficacy in treating autoimmune diseases (e.g., SLE) and that they do not necessarily need to be Tregs.

Response 4
We appreciate this suggestion. We have now clarified that, in most preclinical and translational studies, CAR-/TCR-modified Tregs are generated from natural Tregs (nTregs; CD4⁺CD25⁺CD127low FOXP3⁺) that are isolated and subsequently engineered. We also note that Teff/Tconv can be redirected by FOXP3 transduction in combination with a CAR/eTCR, but this remains a more experimental route. We have additionally added a brief statement summarising the clinical experience with CD19-directed CAR T cells in autoimmune diseases (e.g., SLE)—highlighting that these products are conventional CAR-T cells (not Tregs) and have shown high efficacy in early clinical studies.
Where: Section 6.4.

Updated text inserted (excerpts):
“In most preclinical and translational studies, however, CAR-/TCR-modified Tregs are derived from natural Tregs (CD4⁺CD25⁺CD127^low^ FOXP3⁺) that are isolated and then genetically engineered, whereas Teff-to-Treg conversion via FOXP3 plus a CAR/eTCR remains an alternative, more experimental route” (lines 428-431)

“Separately, it should be noted that CD19-directed CAR-T cells used to treat autoimmune disease are typically conventional T cells rather than Tregs, and multiple early clinical reports (including case series and prospective cohorts) have shown high rates of remission in refractory conditions such as SLE, supporting the concept of B-cell depletion/immune reset without the need for a Treg phenotype” (lines 473-477)

(References supporting both points have been added in the manuscript; Table 3 cross-references the source cell type used in each model.)

We thank the reviewer again for the thoughtful input. We believe these revisions improve clarity and will help readers navigate the field more easily. We remain at the reviewer’s and editor’s disposal for any further suggestions or adjustments.

Reviewer 2 Report

Comments and Suggestions for Authors

This review written by Dr. Francisco Lozano et al group provides a well-written and comprehensive overview of regulatory T-cell (Treg) biology, their mechanisms of action, and therapeutic applications. The review is timely and informative, highlighting key signaling pathways and translational directions in Treg-based immunotherapy. The organization and language are generally clear, and the figures effectively complement the discussion. However, some conceptual areas, particularly the explanation of Treg generation in the thymus need additional clarity, and minor textual and figure-legend corrections are required. Overall, this is a strong manuscript that can be improved with minor revisions.

Comments

  1. In the Introduction, the authors should briefly mention about the 2025 Nobel Prize in Physiology or Medicine, which recognized Mary E. Brunkow, Fred Ramsdell, and Shimon Sakaguchi for elucidating the central role of FOXP3 and regulatory T cells (Tregs) in immune tolerance. A concise paragraph could be included, such as:

“The 2025 Nobel Prize in Physiology or Medicine recognizes Mary E. Brunkow, Fred Ramsdell, and Shimon Sakaguchi for elucidating the central role of FOXP3 and regulatory T cells (Tregs) in the maintenance of immune tolerance. This discovery reframed immune regulation as an active process essential for preventing autoimmunity and maintaining immune balance.”

Please add key references (Nature Genetics 27:68–73 (2001); Science 299:1057–1061 (2003); Nature Immunology 4:330–336 (2003); Immunity 22:329–341 (2005)).

  1. The description of positive selection and generation of Tregs in the thymus is not entirely convincing. The authors should elaborate how Tregs arise through agonist selection rather than simple positive selection, emphasizing the role of TCR affinity, IL-2/STAT5 signaling, and costimulatory cues (CD28, ICOS). A simple figure illustrating the selection thresholds for deletion, Treg, and conventional T-cell fates would strengthen this section.
  2. Figure legends require correction. Several legends are incomplete or contain small formatting issues (e.g., mis-spaced symbols, incorrect abbreviations). Please revise legends for accuracy, consistent gene/protein nomenclature (e.g., Foxp3 for mouse, FOXP3 for human), and clarity of color/key references.
  3. There are minor typographical errors throughout the manuscript (for example, line 590 and in the figure legends). A thorough proofread should resolve these.
  4. The review would benefit from slightly expanded discussion on Treg plasticity and stability under inflammatory conditions, connecting this to current therapeutic approaches.
  5. Overall, this review is well-structured and informative. After these minor scientific and editorial revisions, especially adding the Nobel Prize context and clarifying Treg thymic selection, it will make an excellent and timely contribution.

Author Response

We sincerely thank Reviewer 2 for the careful and constructive assessment of our manuscript. We found the critique coherent, meticulous and clearly aimed at strengthening the work. We have implemented the suggested changes and believe they have improved both the conceptual clarity and editorial quality of the review.

Comment 1

  1. In the Introduction, the authors should briefly mention about the 2025 Nobel Prize in Physiology or Medicine, which recognized Mary E. Brunkow, Fred Ramsdell, and Shimon Sakaguchifor elucidating the central role of FOXP3 and regulatory T cells (Tregs) in immune tolerance. A concise paragraph could be included, such as:

“The 2025 Nobel Prize in Physiology or Medicine recognizes Mary E. Brunkow, Fred Ramsdell, and Shimon Sakaguchi for elucidating the central role of FOXP3 and regulatory T cells (Tregs) in the maintenance of immune tolerance. This discovery reframed immune regulation as an active process essential for preventing autoimmunity and maintaining immune balance.”

Please add key references (Nature Genetics 27:68–73 (2001); Science 299:1057–1061 (2003); Nature Immunology 4:330–336 (2003); Immunity 22:329–341 (2005)).

Response 1: 

We fully agree. Indeed, when we saw the Nobel announcement we considered it remarkably timely in relation to this review. We have now added the requested mention in the Introduction and linked it explicitly to the conceptual basis for Treg-based ACT, incorporating all four key foundational references suggested by the reviewer.
Where: Introduction, lines 52–58 (new text; highlighted).
Updated text (excerpt):
“In recognition of the centrality of regulatory T cells (Tregs) to immune homeostasis, the 2025 Nobel Prize in Physiology or Medicine honoured Mary E. Brunkow, Fred Ramsdell and Shimon Sakaguchi for discoveries that established FOXP3⁺ Tregs as key enforcers of immune tolerance. This milestone crystallised the concept that immune regulation is an active process essential to preventing autoimmunity and maintaining balance, providing the conceptual foundation for the Treg-based adoptive cell therapies discussed in this review.”
References added: Nature Genetics 27:68–73 (2001); Science 299:1057–1061 (2003); Nature Immunology 4:330–336 (2003); Immunity 22:329–341 (2005).

Comment 2

The description of positive selection and generation of Tregs in the thymus is not entirely convincing. The authors should elaborate how Tregs arise through agonist selectionrather than simple positive selection, emphasizing the role of TCR affinity, IL-2/STAT5 signaling, and costimulatory cues (CD28, ICOS). A simple figure illustrating the selection thresholds for deletion, Treg, and conventional T-cell fates would strengthen this section.

Response 2

We agree—our initial text was overly simplified. We have revised Section 3.1 to clarify that thymic Tregs arise via agonist selection, with explicit description of the TCR signal-strength window and the roles of IL-2/STAT5, CD28, ICOS, and medullary antigen provision by AIRE/FEZF2-dependent mTECs and dendritic cells.

Where: Section 3.1, lines 97–118 (revised/expanded; highlighted).
Updated text:
“This lineage commitment is not accomplished by simple positive selection but rather through an agonist-selection programme that depends on relatively strong self-reactive TCR signals delivered in the thymic medulla [45,46]. Within this framework, TCR signal strength operates within a window of outcomes: low/tonic signals support conventional positive selection of Tconv, very high-affinity signals trigger clonal deletion, and intermediate-to-high ‘agonist’ signals—integrated with key cytokine and co-stimulatory cues—divert cells to the Treg lineage [46]. IL-2–STAT5 signalling is indispensable for Foxp3 induction and stabilisation, with STAT5 engaging Foxp3 regulatory elements and scaling the size of the tTreg niche [47–49]. CD28 co-stimulation is required for tTreg development, acting in concert with TCR signals and IL-2 [50], while ICOS–ICOSL interactions fine-tune selection and support Treg differentiation/maintenance in the thymic environment [51,52]. Antigen availability is provided by AIRE- and FEZF2-dependent programmes in medullary thymic epithelial cells (mTECs) and by dendritic cells that acquire and present tissue-restricted antigens; these antigen-presentation axes coordinate deletion versus Treg diversion of strongly self-reactive clones, shaping the emergent Treg repertoire and tissue specificity [53–55]. Collectively, these data establish agonist selection—rather than simple positive selection—as the defining pathway for thymic Treg generation, integrating TCR signal strength with IL-2/STAT5 and CD28/ICOS co-stimulation in the context of medullary antigen presentation [45–47,56]. Together, these convergent signals initiate FOXP3 transcription and the early stabilisation of the Treg lineage programme, providing a mechanistic bridge from thymic selection to lineage commitment.”

Comment 3

Figure legends require correction. Several legends are incomplete or contain small formatting issues (e.g., mis-spaced symbols, incorrect abbreviations). Please revise legends for accuracy, consistent gene/protein nomenclature (e.g., Foxp3for mouse, FOXP3 for human), and clarity of color/key references.

Response 3

Thank you for this observation. We have proofread all figure legends and made targeted edits to improve consistency and clarity—harmonising FOXP3/Foxp3 and marker notation (e.g., CD4⁺, CD25high, CD127low), correcting spacing/hyphenation, defining abbreviations at first mention where needed, and clarifying colour/key references. These adjustments are highlighted in the revised file.
Where: Figure legends 1–4 (updated; highlighted).

Comment 4

There are minor typographical errors throughout the manuscript (for example, line 590 and in the figure legends). A thorough proofread should resolve these.

Response 4

We performed a thorough proofread and corrected typographical issues across the manuscript and legends.

Comment 5 

The review would benefit from slightly expanded discussion on Treg plasticity and stability under inflammatory conditions, connecting this to current therapeutic approaches.

We agree and have expanded the biology section and connected these concepts to therapeutic design considerations.

Where (biology): Section 3.1, lines 131–147 (highlighted).

Updated text (excerpt):

“Although FOXP3/Foxp3 expression and the Treg epigenome normally confer lineage stability, inflammatory cytokines and strong co-stimulation can bias Treg towards functional plasticity… IL-6–STAT3 signalling antagonises Treg programmes and favours Th17-like features… At the molecular level, FOXP3 cis-regulatory elements, notably CNS2/TSDR, are crucial to maintain high FOXP3 expression and lineage stability… targeted TSDR demethylation stabilises FOXP3 in human T cells; conversely, loss of CNS2 function destabilises the lineage… Beyond FOXP3, additional transcription factors… and pathways (PI3K–AKT–mTOR) modulate the balance between suppressive and effector programmes [58–65].”

Where (applications): Section 7.3, lines 672–688 (new paragraph; highlighted).

Updated text (excerpt):

“Insights into Treg plasticity under inflammatory conditions argue for product and regimen designs that actively promote FOXP3 stability and resist Th1/Th17 deviation. First, source selection matters: CD45RA⁺ (naïve) nTregs display superior epigenetic stability… [219,220]. Second, ex vivo culture conditions can be tuned to preserve lineage commitment: IL-2–STAT5 support together with mTOR inhibition (rapamycin) maintains FOXP3… [220,221]. Approaches that favour FOXP3-CNS2/TSDR demethylation… further consolidate lineage stability [222,223]. Third, co-interventions in vivo… low-dose IL-2… IL-6/STAT3 pathway blockade… [224–226]. Taken together, stability-aware sourcing, manufacturing and adjunct therapy can be systematically combined to increase the odds of durable Treg engraftment, function and clinical benefit in autoimmune settings.”

We appreciate the reviewer’s insightful feedback and positive assessment. We hope these revisions satisfactorily address all comments. We remain at the reviewer’s and editor’s disposal for any additional suggestions or refinements.